# Autoinhibition of ankyrin-B/G membrane target bindings by intrinsically disordered segments from the tail regions

Keyu Chen[1†], Jianchao Li[1†], Chao Wang[1,2], Zhiyi Wei[1,3], Mingjie Zhang[1,4*]

[1]Division of Life Science, State Key Laboratory of Molecular Neuroscience, Hong Kong University of Science and Technology, Hong Kong, China; [2]School of Life Sciences, University of Science and Technology of China, Hefei, Anhui, China; [3]Department of Biology, South University of Science and Technology of China, Shenzhen, China; [4]Center of Systems Biology and Human Health, Institute for Advanced Study, Hong Kong University of Science and Technology, Hong Kong, China

**Abstract** Ankyrins together with their spectrin partners are the master organizers of micron-scale membrane domains in diverse tissues. The 24 ankyrin (ANK) repeats of ankyrins bind to numerous membrane proteins, linking them to spectrin-based cytoskeletons at specific membrane microdomains. The accessibility of the target binding groove of ANK repeats must be regulated to achieve spatially defined functions of ankyrins/target complexes in different tissues, though little is known in this regard. Here we systemically investigated the autoinhibition mechanism of ankyrin-B/G by combined biochemical, biophysical and structural biology approaches. We discovered that the entire ANK repeats are inhibited by combinatorial and quasi-independent bindings of multiple disordered segments located in the ankyrin-B/G linkers and tails, suggesting a mechanistic basis for differential regulations of membrane target bindings by ankyrins. In addition to elucidating the autoinhibition mechanisms of ankyrins, our study may also shed light on regulations on target bindings by other long repeat-containing proteins.

DOI: https://doi.org/10.7554/eLife.29150.001

*For correspondence:
mzhang@ust.hk

[†]These authors contributed equally to this work

## Introduction

Ankyrins are a widely expressed scaffold protein family, which mainly function to link great varieties of functionally related but structurally diverse integral membrane proteins to the spectrin-based cytoskeletons (*Bennett and Baines, 2001*; *Bennett and Healy, 2009*; *Bennett and Lorenzo, 2013*). Mutations of ankyrins are known to be associated with various diseases in humans such as hereditary spherocytosis (*Eber et al., 1996*; *Lux et al., 1990*; *Gallagher, 2005*), cardiac arrhythmia syndromes and sinus node dysfunction (*Mohler et al., 2003*; *Mohler et al., 2007*; *Le Scouarnec et al., 2008*; *Hashemi et al., 2009*), bipolar disorders (*Baum et al., 2008*; *Ferreira et al., 2008*; *Schulze et al., 2009*; *Scott et al., 2009*; *Takata et al., 2011*; *Dedman et al., 2012*; *Rueckert et al., 2013*), schizophrenia (*Nie et al., 2015*; *Ripke et al., 2011*) and autism spectrum disorder (*Willsey et al., 2013*; *Parikshak et al., 2013*; *Shi et al., 2013*; *Iqbal et al., 2013*). In vertebrates, the ankyrin family consists of three members: ankyrin-R (AnkR), ankyrin-B (AnkB) and ankyrin-G (AnkG). They share similar domain organizations (*Figure 1A*), but usually locate at different subcellular regions and perform divergent physiological functions (*Mohler et al., 2002*; *Abdi et al., 2006*; *He et al., 2013*; *Bennett and Lorenzo, 2016*).

Each ankyrin contains an N-terminal membrane binding domain (MBD), which is composed of 24 ANK repeats and responsible for binding to diverse membrane targets. The 24 ANK repeats of

**eLife digest** The membrane that surrounds the cells of animals is organized into regions known as microdomains. Different microdomains have different roles; for example, one microdomain may strengthen the cell while another may contain the apparatus used by cells to signal to each other.

A family of proteins called the ankyrins plays a central role in forming microdomains, and mutations to these proteins have been linked to many diseases, including heart rhythm abnormalities and bipolar disorder. Ankyrins connect certain proteins in the cell membrane to each other and to the network of microfilaments that organize the cell interior. A region called the membrane binding domain at one end of the ankyrin interacts with the membrane proteins. The other end, or "tail", of the ankyrin is thought to regulate this interaction, although little is known about how it does so.

The members of the ankyrin family are similar but tend to localize to different parts of the cell and play different roles. Chen, Li et al. have now used biochemical and structural techniques to analyze the sequences of two ankyrins, called Ankyrin-B and Ankyrin-G. The results show that the tail regions of both of these proteins contain three "autoinhibition" segments that restrict the activity of the ankyrins. Each segment independently interacts with different sites on the ankyrin's membrane binding domain, and this interaction joins the ends of the protein together into a 'head-to-tail' conformation that prevents it binding to membrane proteins. The autoinhibition segments of Ankyrin-B and Ankyrin-G are quite different, which Chen, Li et al. suggest could explain why these proteins have different activities.

Ankyrins play a vital role in many tissues, and so the findings presented by Chen, Li et al. will be of interest to scientists working in a wide range of basic biology and medical research fields. An immediate question for further investigation is how the autoinhibition of ankyrins is regulated to allow ankyrin-organized microdomains to form and work properly. Results from such studies will help researchers to understand the mechanisms that underlie diseases caused by mutations of ankyrins or the proteins that they bind to, and to develop potential therapies for these diseases.
DOI: https://doi.org/10.7554/eLife.29150.002

ankyrins are among the longest extended repeat-containing domains in living organisms. A trademark of long repeat-containing proteins is their formations of long solenoid structures with high conformational plasticity (*Lee et al., 2006b*; *Wang et al., 2014*). Following the MBD is an unstructured linker region of ~130 amino acid residues, followed sequentially by a ZU5-ZU5-UPA supramodule responsible for spectrin binding (SBD) (*Mohler et al., 2004a*; *Wang et al., 2012*), a death domain (DD), and an intrinsically disordered C-terminal tail (CT) with varying lengths (*Figure 1A*). The folded structural domains (i.e. MBD, SBD and DD) in ankyrins share very high sequence similarities (*Figure 1A*). In contrast, the unstructured regions of ankyrins (i.e. the linker region and CT), although fairly conserved within each isoform, are more divergent among the family members (*Figure 1A*). Presumably, these isoform-specific and unstructured linker and tail regions are largely responsible for distinct functions of each member of the ankyrin family proteins (see *Abdi et al., 2006* and *He et al., 2013* for examples). However, very little is known regarding how the unique linker and tail regions specifically modulate functions of each ankyrin.

Ankyrin is an ancient molecule that appeared as early as in bilaterians over 500 million years ago (*Cai and Zhang, 2006*; *Bennett and Lorenzo, 2013*). The 24 ANK repeats of ankyrins have retained a remarkable level of amino acid sequence conservation throughout the evolution, and different ANK repeats binding membrane proteins with diverse sequences have evolved at different stages of the evolution (*Hill et al., 2008*; *Bennett and Lorenzo, 2013*; *Bennett and Lorenzo, 2016*). In an earlier study, we demonstrated that the highly conserved 24 ANK repeats can form multiple semi-independent target binding sites, and each capable of recognizing short and intrinsically disordered segments with variable amino acid sequences distributed on target proteins (*Wang et al., 2014*). As such, ANK repeats are capable of binding to a very diverse array of target proteins with high specificity, a strategy that may be commonly employed by long repeat-containing scaffold proteins (*Wang et al., 2014*; *Chook and Blobel, 2001*; *Conti et al., 1998*; *Graham et al., 2000*; *Huber and Weis, 2001*; *Kobe, 1999*; *Xu et al., 2010*; *Zhu et al., 2011*). The target bindings of ankyrin's ANK

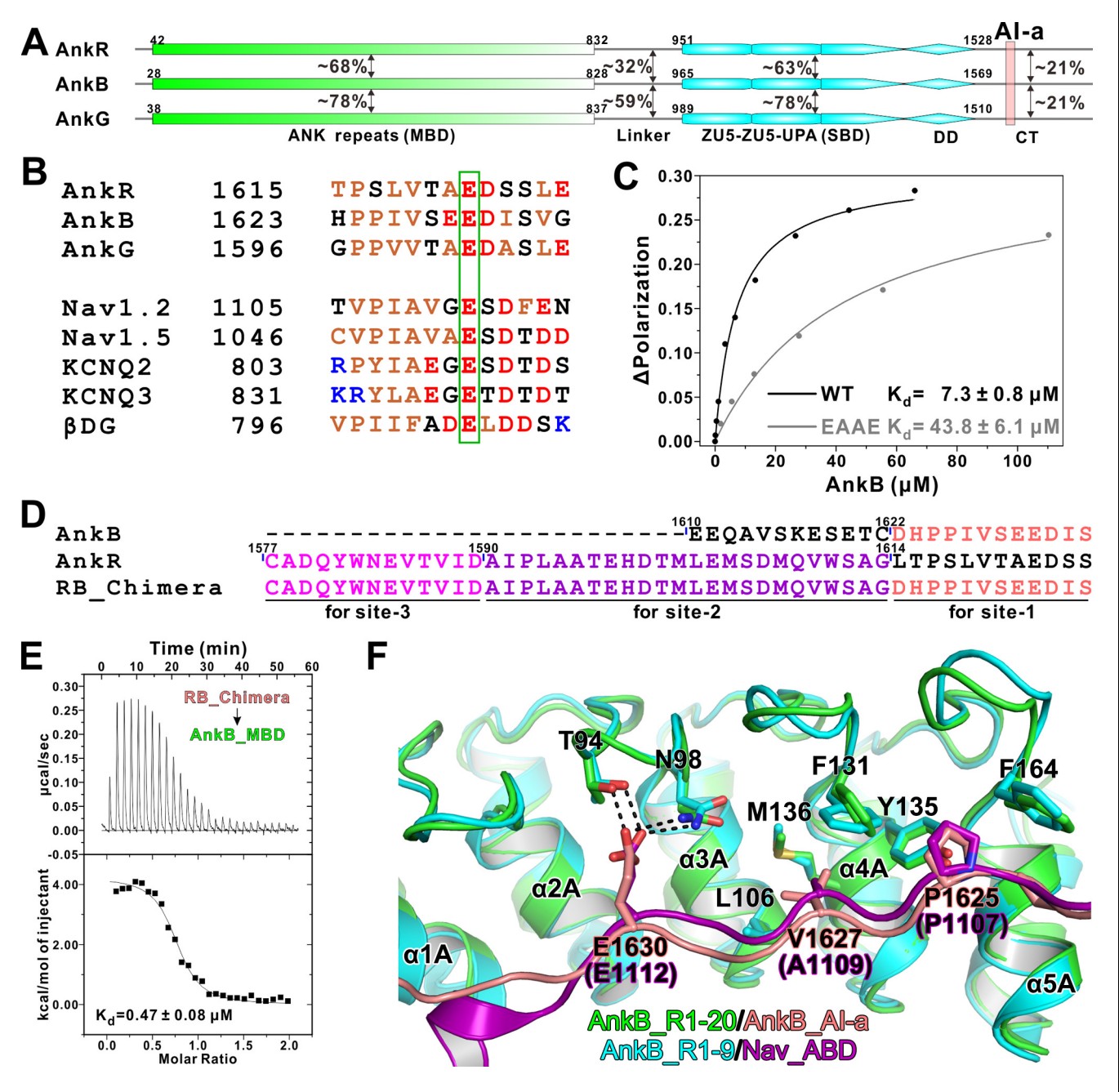

**Figure 1.** Biochemical and structural characterization of the AI-a/MBD interaction. (**A**) Schematic diagram showing the domain organizations of AnkR/B/G and their amino acid sequence identities. (**B**) Amino acid sequence alignments of AI-a from AnkB/G with several MBD 'site-1' binding sequences from AnkR CT or other targets. The critical Glu residues are highlighted with a green box. (**C**) Fluorescence polarization-based measurement of the binding affinity between AnkB_AI-a and AnkB_MBD or its R1 charge reversal mutant ('EAAE'). (**D**) The design of AnkR/B_Chimera construct for crystallization. The AnkB_AI-a sequence (colored in salmon) was fused to the C-terminus of 'site-2, 3' binding AnkR_CT (colored in purple and magenta respectively). (**E**) ITC result showing the strong interaction between AnkR/B_Chimera and MBD. (**F**) Structural comparison of the MBD 'site-1' bindings of AnkB_AI-a (colored in green and salmon) and Nav_ABD (colored in cyan and purple) showing that the two bindings essentially share the same mode. Residues critical for the interaction were highlighted with stick models. Hydrogen bonds were indicated with dashed lines (the same labeling method is used throughout the manuscript for all structural figures).

DOI: https://doi.org/10.7554/eLife.29150.003

repeats must be regulated during the biogenesis and trafficking processes of ankyrins and their targets, as otherwise functionally unrelated membrane targets would be misdirected to the same sites in cellular membranes. Additionally, the formation of each ankyrin/target complex (e.g. the ankyrin-B/G and sodium channel complexes) at specific membrane domains also needs to be regulated to fulfill excitation/resting cycles of excitable tissues such as neurons and muscles (*Bréchet et al., 2008*; *Garver et al., 1997*; *Whittard et al., 2006*). Earlier studies have suggested that ankyrins can adopt autoinhibited conformation (i.e. by using sequences outside the ANK repeats to directly interact with the repeats) to regulate their target bindings (*Abdi et al., 2006*; *Davis et al., 1992*; *He et al., 2013*; *Wang et al., 2014*), but the mechanisms of the autoinhibition are unknown.

In this study, we systematically investigated the autoinhibition mechanisms of ANK repeats of ankyrin-B and G (AnkB and G). We identified three distinct autoinhibitory segments from the intrinsically disordered linker and CT regions of AnkB, and elucidated the inhibition mechanism of these segments by detailed biochemical and structural investigations. Our study reveals that the three inhibitory segments spread along the entire 24 ANK repeats and bind to the repeats in a quasi-independent manner, so that the inhibitory sites may be combinatorially regulated in response to different binding targets. We further demonstrated that AnkG also adopts a similar overall autoinhibition strategy as AnkB does, but with its own unique binding features. Collectively, the findings of our study not only provide a framework for understanding the interactions and regulations of ankyrins with their membrane targets, but may also be helpful in understanding the regulation of target recognitions by other long repeat-containing scaffold proteins in general.

## Results

### A segment in AnkB C-terminal tail binds to the N-terminal of ANK repeats

Ankyrin MBD can be inhibited by its CT regulatory domain in both AnkR and AnkB (*Abdi et al., 2006*; *Davis et al., 1992*), and previously we have biochemically and structurally characterized the intramolecular interaction between the 48-residue AnkR CT regulatory domain with all three MBDs from the ankyrin family members (*Wang et al., 2014*). Moreover, our analysis illustrated that the MBDs of AnkR/B/G are highly conserved and share essentially the same binding properties to diverse binding partners (*Wang et al., 2014*). In the structure of AnkB_MBD in complex with AnkR_CT, AnkR_CT adopts an extended conformation lining the inner groove of ANK repeats 1 ~ 14 of AnkB_MBD (*Wang et al., 2014*). Although AnkB and G do not contain AnkR_CT like sequences in their tails, more thorough sequence analysis revealed that both AnkB and G contain a 13-residue segment (denoted as 'AI-a' for the AutoInhibition segment-a, *Figure 1A*) that share a similar sequence pattern with the last 13-residue fragment of AnkR_CT and several MBD 'site-1' binding targets such as $Na^+$ and $K^+$-channels (*Figure 1B*). This 13-resdiue AI-a fragment is almost completely conserved in AnkB and AnkG among vertebrates (alignments not shown). Importantly, a Glu residue corresponding to Glu1622 in AnkR_CT and Glu1112 in Nav1.2, which has been shown to be absolutely required for AnkR_CT and Nav1.2 to bind to ANK repeats (*Mohler et al., 2004b*; *Wang et al., 2014*), is also invariant in AnkB and G (*Figure 1B*). We found that a peptide encompassing this AnkB_AI-a segment binds to AnkB ANK repeats with a dissociation constant (Kd) ~7.3 μM based on a fluorescence polarization binding assay (*Figure 1C*). Point mutations of the positively charged residues in the repeat 1 (R1) of ANK repeats ($^{37}$RAAR$^{40}$ to $^{37}$EAAE$^{40}$) of AnkB significantly decreased its binding to the AI-a peptide (*Figure 1C*), consistent with our previous structural analysis that the positively charged residues in the ANK repeat R1 play a role in binding to the negatively charged residues from the 'site-1'-binding peptides.

We attempted to elucidate the molecular basis governing the AnkB ANK repeats/AI-a interaction by solving the complex structure using a similar fusion strategy as we used earlier for the Nav1.2 peptide (*Wang et al., 2014*); i.e. by fusing ANK repeats R1-20 to the C-terminal tail of AnkB AI-a. However, this effort was not successful. As an alternative approach, we replaced the corresponding sequence of AnkR_CT with that of AnkB_AI-a and produced an AnkR/B_CT Chimera (*Figure 1D*). Isothermal titration calorimetry (ITC) experiments detected a strong interaction between this chimera peptide and AnkB_MBD, with a Kd value comparable to that of AnkR_CT with AnkB_MBD (*Figure 1E*). This chimera peptide was fused to the N-terminus of AnkB_MBD for crystallization trials

as we have demonstrated for AnkR_CT. With this strategy, we successfully obtained crystals of the AnkR/B_CT Chimera/AnkB_repeats_R1-20 fusion protein, and the crystals were diffracted up to 3.3 Å. The structure of the fusion protein was determined by the molecular replacement method using the AnkB_repeats_R1-24 structure as the model (*Wang et al., 2014*) (*Table 1*). Consistent with the biochemical analysis, the structure showed that the AnkB_AI-a segment binds to 'site-1' of AnkB_MBD (repeats_R1-5) using essentially the same binding mode as we observed in the AnkR_CT/AnkB_MBD and Nav1.2_ABD/AnkB_R1-9 complex structures (*Wang et al., 2014*) (*Figure 1F*). In particular, Glu1630 from AnkB_AI-a occupies the identical position as Glu1622 in AnkR or Glu1112 in Nav1.2 on AnkB_R1-5, by forming strong hydrogen bonds with Thr94 and Asn98 in the R2-R3 finger loop (*Figure 1F*). Additionally, hydrophobic interactions between 'PPIV'

**Table 1.** Statistics of X-ray Crystallographic Data Collection and Model refinement

Data collection

| Data sets | RB-Chimera/AnkB_R1-20 | AI-b/AnkB_R8-M14 | AI-c/AnkB_R13-24 |
|---|---|---|---|
| Space group | $R32$ | $P6_522$ | $P2_12_12_1$ |
| Wavelength (Å) | 0.979 | 0.979 | 0.979 |
| Unit Cell parameters (Å) | a = b = 179.79, c = 227.10 $\alpha=\beta=90°$, $\gamma = 120°$ | a = b = 186.09, c = 75.35 $\alpha=\beta=90°$, $\gamma = 120°$ | a = 29.30, b = 127.80, c = 257.55 $\alpha=\beta=\gamma=90°$ |
| Resolution range (Å) | 50–3.3 (3.36–3.30) | 50–2.35 (2.39–2.35) | 50–1.95 (1.98–1.95) |
| No. of unique reflections | 20949 (1025) | 31971 (1562) | 68019 (3163) |
| Redundancy | 5.3 (5.5) | 4.1 (3.8) | 3.1 (3.2) |
| I/σ | 25.5 (3.3) | 17.0 (2.0) | 22.8 (2.4) |
| Completeness (%) | 97.6 (98.9) | 97.5 (97.9) | 94.4 (92.2) |
| $R_{merge}$[*] (%) | 10.3 (72.1) | 11.4 (89.3) | 8.2 (70.3) |
| Structure refinement | | | |
| Resolution (Å) | 50–3.3 (3.42–3.30) | 50–2.35 (2.42–2.35) | 50–1.95 (2.02–1.95) |
| $R_{cryst}$[†]/$R_{free}$[‡] (%) | 18.27/22.96 (25.40/31.06) | 19.21/23.25 (24.28/28.57) | 18.38/21.98 (24.40/29.34) |
| Rmsd bonds (Å)/angles (°) | 0.007/1.011 | 0.008/0.955 | 0.009/0.996 |
| Average B factor [§] | 102.8 | 46.3 | 30.9 |
| No. of atoms | | | |
| Protein atoms | 5019 | 3421 | 6128 |
| Water | 0 | 64 | 274 |
| Other molecules | 50 | 102 | 5 |
| No. of reflections | | | |
| Working set | 18953 | 30384 | 64519 |
| Test set | 1980 | 1540 | 3357 |
| Ramachandran plot regions [§] | | | |
| Favored (%) | 95.2 | 98.0 | 98.7 |
| Allowed (%) | 4.8 | 2.0 | 1.3 |
| Outliers (%) | 0 | 0 | 0 |

Numbers in parentheses represent the value for the highest resolution shell.

\* $R_{merge} = \Sigma |I_i - <I>| / \Sigma I_i$, where $I_i$ is the intensity of measured reflection and $<I>$ is the mean intensity of all symmetry-related reflections.

b† $R_{cryst}=\Sigma||F_{calc}| - |F_{obs}||/\Sigma F_{obs}$, where $F_{obs}$ and $F_{calc}$ are observed and calculated structure factors.

c‡ $R_{free}= \Sigma_T||F_{calc}| - |F_{obs}||/\Sigma F_{obs}$, where T is a test data set of about 5% or 10% of the total unique reflections randomly chosen and set aside prior to refinement.

d§ B factors and Ramachandran plot statistics are calculated using MOLPROBITY (*Chen et al., 2010*).

DOI: https://doi.org/10.7554/eLife.29150.004

from AI-a and hydrophobic residues (including the two critical Phe residues, Phe131 and Phe164, from R4 and R5; see *Wang et al., 2014*) from R3-R5 also contribute to the binding (*Figure 1F*).

## Two discrete segments from the linker connecting AnkB MBD and SBD bind to the middle and C-terminal parts of ANK repeats respectively

Though the conservation among the three isoforms is limited, the amino acid sequences of the linker connecting MBD and SBD within each isoform of ankyrins are highly conserved (*Figure 1A*). The crystal structure of AnkR MBD C-terminal 12 ANK repeats (PDB ID: 1N11, referred to as AnkR_C12) contains a 14-residue fragment of the linker that folds back and binds to the last five repeats (R20-24) of MBD (*Michaely et al., 2002*) (also see below for more details). Recently, the linker region of AnkB was reported to directly interact with MBD, thus preventing AnkB from localizing to plasma membranes (*He et al., 2013*). These studies suggest that, besides the AI-a in their CT, the linker regions of ankyrins also play important autoinhibitory roles in regulating functions of MBDs.

We chose AnkB to investigate the detailed mechanism governing the linker region-mediated autoinhibition of ankyrins. To probe the linker-mediated autoinhibition, we used three AnkB_MBD binding targets (Nav1.2, E-cadherin and NF186/L1CAM), each with distinct MBD binding mode (see [*Wang et al., 2014*]; and our unpublished data on E-cadherin), to test their bindings to three versions of extended MBD: one containing a short 20-residue linker (aa 28–847, roughly corresponding to the above-mentioned AnkR_C12 structure but with additional six residues. See *Table 2* for a list of key constructs used in the study), one containing the entire linker (aa 28–965), and the third one with the entire linker as well as the entire CT regulatory domain (i.e. the full-length AnkB with the spectrin binding ZZUD tandem deleted, denoted as AnkB ΔZZUD) (*Figure 2A*). Interestingly, extending the linker region from residue 847 to 965 (AnkB 28–847 *vs* AnkB 28–965) invariably led to weakened or completely loss of interactions between MBD and the three target proteins (*Figure 2B and C*), indicating that an additional segment within the residues 848–965 can bind to the ANK repeats of AnkB and acts as another autoinhibitory sequence. Including the AI-a fragment in the MBD extension (AnkB ΔZZUD) further eliminated the remaining binding of Nav1.2 (*Figure 2B and C*), and this is consistent with our findings that AI-a and Nav1.2 bind to R1-5 of MBD in a mutually competitive manner (*Figure 1F*; and *Wang et al., 2014*).

We confirmed the direct interaction between the entire linker (aa 828–965) with the AnkB ANK repeats only (aa 28–827) using an ITC-based binding assay, and data showed a strong binding (Kd about 0.044 μM, *Figure 2D*). The entire linker (aa 828–965) was found to bind to AnkB 28–847 with a Kd ~4.2 μM (*Figure 2E*), further indicating that a fragment within residues 848–965 can directly bind to AnkB_MBD. Another version of extended MBD, AnkB 28–873 interacts with the entire linker (aa 828–965) with a similar affinity as AnkB 28–847 does (*Figure 2F*), suggesting that residue 848–873 has marginal impact on the AnkB autoinhibition. Taken together, the above biochemical mapping experiments suggested that the AnkB linker region contains two discrete autoinhibitory

**Table 2.** Constructs of ankyrins used in this study

|  | AnkB | AnkG |
| --- | --- | --- |
| AI-a | 1623–1635 | 1596–1608 |
| AI-b | 865–896 | 875–903 |
| AI-c | 828–852 | 837–858 |
| Linker (contains AI-c + AI b) | 828–965 | |
| AI-c + AI-b | 828–896 | 837–920 |
| Linker ΔAI-c | | 861–920 |
| MBD + partial AI-c | 28–847 | 38–855 |
| MBD + full AI-c | 28–873 | |
| MBD + AI c+AI-b | 28–965 | 38–903 |
| ΔZZUD | | |
| MBD | 28–827 | 38–837 |

DOI: https://doi.org/10.7554/eLife.29150.006

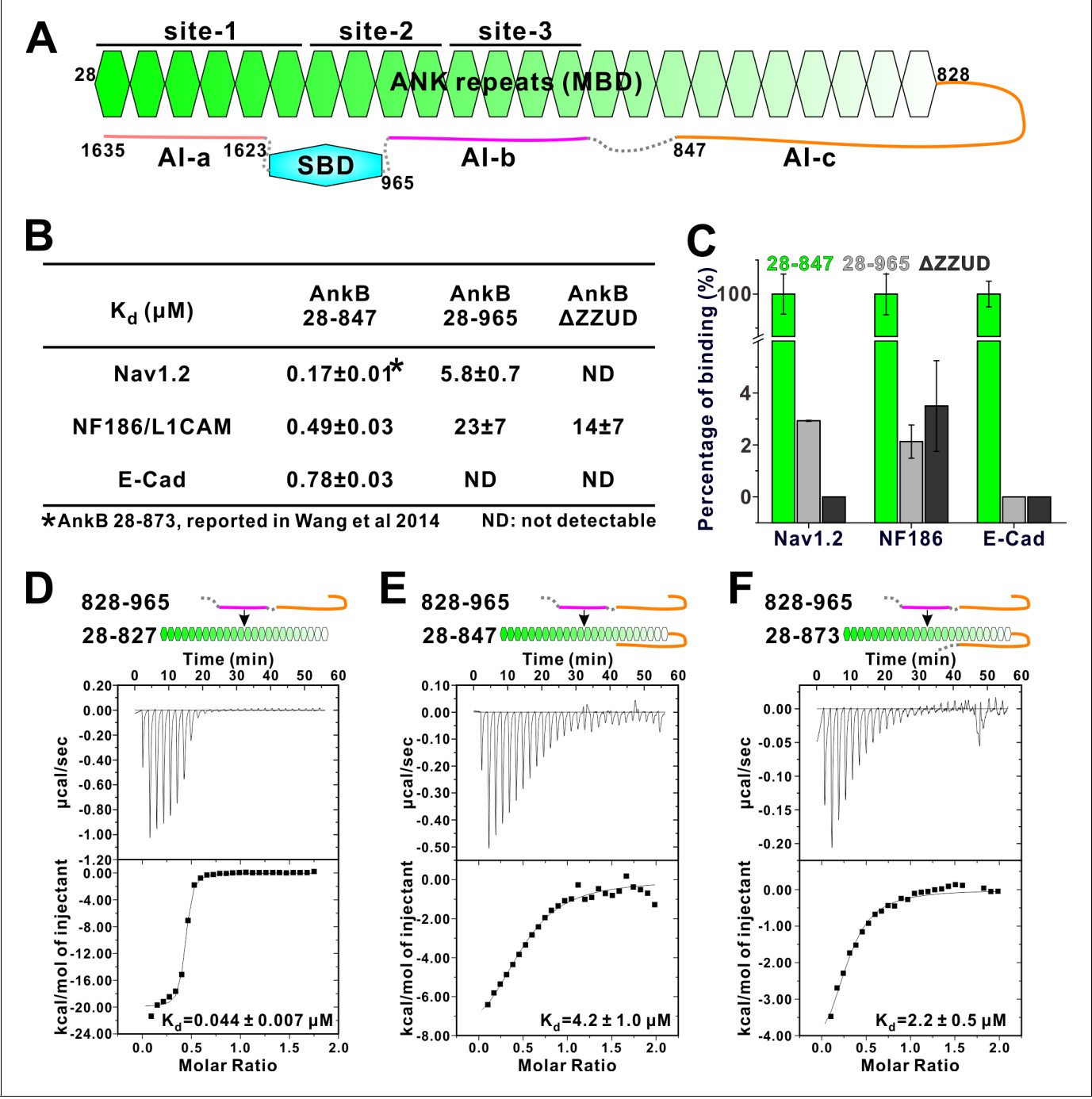

**Figure 2.** Two discrete segments in the AnkB linker region bind to MBD and inhibit its target binding. (**A**) Schematic diagram showing the three autoinhibitory segments (AI-a, b, c) located at the linker and CT regions of AnkB. (**B**) ITC derived binding affinities showing that including longer linker region or the AI-a segment to the AnkB MBD weakened its bindings to targets including Nav1.2, NF186/L1CAM, and E-cadherin. (**C**) Bar graph showing the levels of target binding decreases resulted by the autoinhibitory segments based on the binding data in *Panel B*. (**D–F**) ITC profiles showing direct interactions between the entire linker region of AnkB and different versions of AnkB MBD (D: 28–827, no linker, measured in buffer containing 500 mM NaCl due to poor quality of this protein in 100 mM NaCl buffer; E: 28–847, short linker roughly comparable to the AnkR_C12 structure; F: 28–873, longer linker containing the entire AI-c).

DOI: https://doi.org/10.7554/eLife.29150.005

segments: one segment within the region 848–965 binding to middle repeats of AnkB_MBD (denoted as AI-b), and the other within residues 828–847 overlapping with the short 14-residue segment observed in the AnkR_C12 structure (denoted as AI-c) (*Figure 2A*; and see below for further detailed mapping).

## Structural basis of the AnkB AI-b/ANK repeats interaction

To delineate the mechanism governing the AI-b/MBD interaction, we set out to solve the complex structure. First, we mapped the minimal regions of AI-b and the corresponding repeats from MBD responsible for the interaction. Using the scheme described in *Figure 2E* as the assay, we mapped

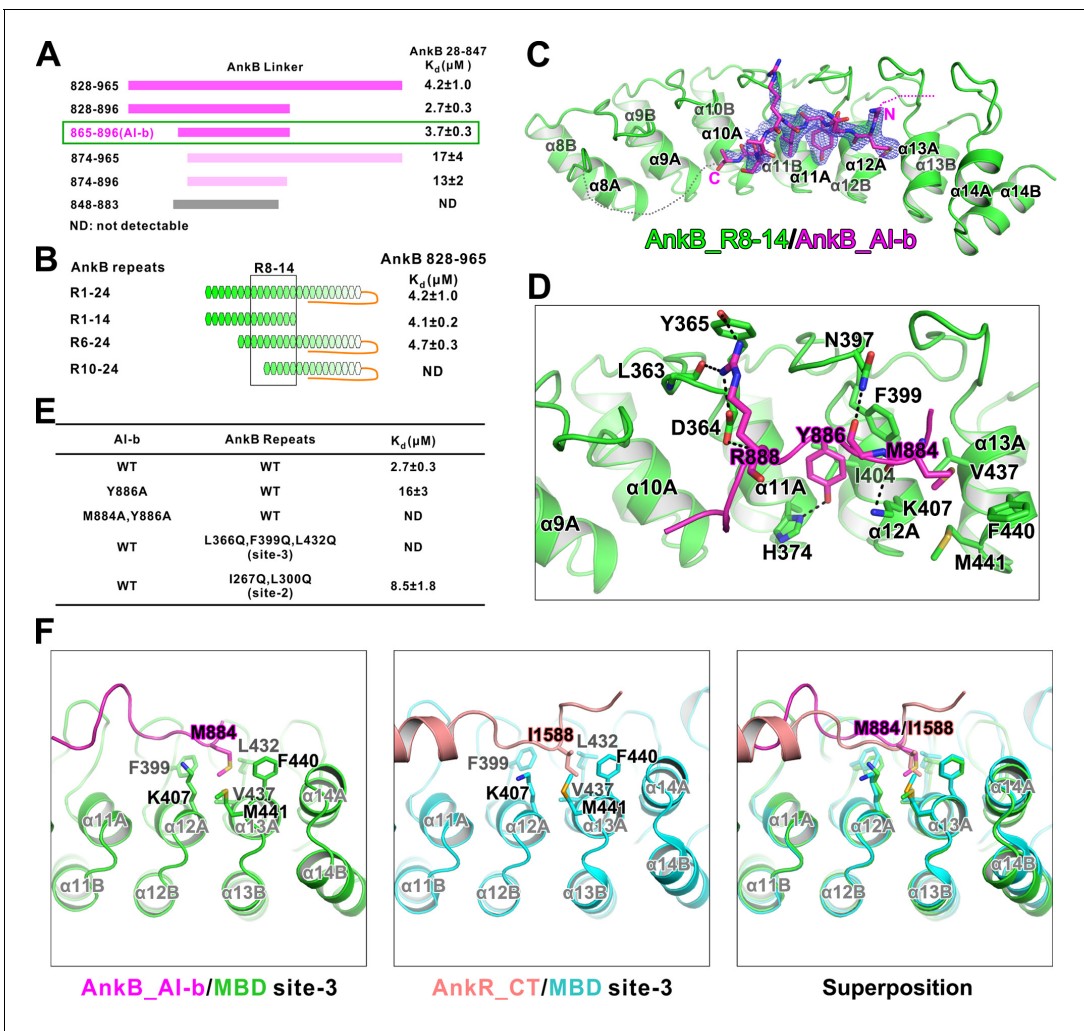

**Figure 3.** Interaction between AnkB_AI-b and AnkB_MBD. (A) ITC-based mapping of the AnkB AI-b region. The minimal region of AI-b is indicated with a green box. The fragments with weakened or complete-loss of binding to MBD were shown in pink and grey bars, respectively. 'ND' represents no detectable binding in the ITC measurement and is used throughout the manuscript. (B) Similar mapping of the minimal ANK repeats responsible for AI-b binding. The repeats used for crystallization (R8-14) are highlighted with a black box. (C) The overall structure and an omit map showing the binding of AnkB_AI-b to AnkB_R8-14. The Fo-Fc density map was generated by deleting the AI-b part from the final model and contoured at 3.0 σ. The AI-b fitting the electron density is displayed in the stick model. (D) Detailed interactions between AnkB_AI-b and AnkB_MBD. The side chains or main chains of the residues involved in the interactions are highlighted in the stick model. Charge-charge and hydrogen bonding interactions are highlighted by dashed lines. (E) ITC derived dissociation constants showing mutations of critical residues weaken or abolish the binding. (F) Structural comparison showing the common hydrophobic pocket in MBD site-3 for binding of AnkB_AI-b and AnkR_CT. Met884 in AnkB_AI-b and Ile1588 in AnkR_CT as well as the residues forming the 'site-3' hydrophobic pocket are shown in stick model. *Left*: AnkB_AI-b/AnkB_MBD, *middle*: AnkR_CT/AnkB_MBD, *right*: superposition of these two.

DOI: https://doi.org/10.7554/eLife.29150.007

AI-b to a 32-aa fragment (aa 865–896) from the linker (*Figure 3A*). Using a similar approach combined with truncations of various repeats, we mapped the AI-b binding regions to R6-14 of MBD (*Figure 3B*).

We determined the crystal structure of AnkB repeats 8–14 fused to the C-terminus of AnkB_AI-b (aa 857–896) at 2.35 Å resolution (*Table 1*). The AI-b peptide binds to the inner groove formed by repeats 10–13 (*Figure 3C*), covering a large portion of the target binding 'site-3' (repeats 11–14) and a small portion of 'site-2' (repeats 7–10) that were defined in our previous study (*Wang et al., 2014*). The structure is also consistent with our biochemical data that the autoinhibitory interaction mediated by AI-b partially inhibits the binding of Nav1.2 and NF186 to MBD (*Figure 2B*), as both targets use the 'site-3' of MBD as one of the binding sites (*Wang et al., 2014*). Consistent with this observation, substitutions of hydrophobic residues within the 'site-3' with Gln (Leu366Gln, Phe399Gln, and Leu432Gln) completely disrupted AI-b's binding to MBD, and substitutions of hydrophobic residues within the 'site-2' with Gln (Ile267Gln, Leu300Gln) mildly weakened the AI-b/MBD interaction (*Figure 3E*; see *Wang et al., 2014* for the rational of the mutations).

The electron densities of the MBD_R8-14 bound AI-b only allowed us to trace residues spanning Gly883-Glu892 (*Figure 3C*). Although residues Glu865-Asp882 could not be traced in the crystal structure, this segment also directly participates in binding to MBD_R8-14, as deletion of residues from 865 to 873 significantly weakened AI-b's binding to AnkB_MBD (*Figure 3A*). The structure of the MBD_R8-14/AI-b complex reveals their binding details: Tyr886 forms a hydrogen bond with His374 and hydrophobic interactions with Phe399 and Ile404; Met884 inserts into a hydrophobic pocket formed by Lys407, Val437, Phe440 and Met441; and Arg888 forms a number of hydrogen bonds with the sidechain of Tyr365 and backbone of Leu363 as well as charge-charge interaction with Asp364 (*Figure 3D*). ITC results showed that single point substitution of Tyr886 from AI-b with Ala significantly decreased its binding to MBD, and Met884Ala, Tyr886Ala double mutations totally abolished the AI-b's autoinhibition (*Figure 3E*). It is noted that Met884 in AI-b occupies the same hydrophobic pocket as Ile1588 in AnkR_CT does, although the two 'site-3' binding fragments share very little amino acid sequence similarity (*Figure 3F*), further highlighting the remarkable capacity of ANK repeats in binding to targets with diverse amino acid sequences (*Wang et al., 2014*).

## Biochemical and structural characterization of the AnkB AI-c/ANK repeats interaction

The AnkR_C12 structure contains a 14-residue linker region (aa 832–845 in mouse AnkR), and 11 residues (aa 835–845) are defined in the folded back structure (*Michaely et al., 2002*). A longer region of the linker (i.e. 828–849 in human AnkB) is highly conserved in all three isoforms of ankyrins (Figure 6A). We speculated that the autoinhibitory AI-c segment of ankyrins may be longer than the 11-residue fragment seen in the structure of AnkR_C12. To ensure that we do not miss any possible residues forming the AI-c binding segment, we prepared a C-terminal 12 ANK repeats of AnkB followed by a 46-residue linker (aa 430–873), which covers a small part of AI-b (*Figure 3A*), and solved the crystal structure of the protein at 1.95 Å resolution (*Table 1*). The overall structure of our AnkB C-terminal 12 repeats-linker is very similar to that of AnkR_C12 (RMSD of 1.04 Å, *Figure 4A*). However, our structure contains a longer linker folded back binding to the AnkB ANK repeats. A 20-residue linker (from Thr828 to Asp847 instead of 11 residues in AnkR_C12) is clearly defined, and this 20-residue linker extends to the repeat R18 (*Figure 4A*). Moreover, we observed a prominently positive-charged surface in the inner groove of R16-18 right next to Asp847, and the five residues following Asp847 are highly negatively charged and conserved ($^{847}$DEEGDD$^{852}$) (*Figure 4B*). We anticipate that this stretch of negatively charged residues is part of AI-c and involved in binding to MBD (likely R16-17 based on the length of the segment) (*Figure 4B*). This idea is supported by the data from an earlier study showing that substitution of '$^{848}$EEGDDT$^{853}$' with 'NAAIRS' in AnkB led to increased membrane localization presumably due to the mutation-induced weakened autoinhibition (*He et al., 2013*). In the AI-c folded-back structure, residues from AI-c form extensive hydrogen bonds with the residues in finger loops of ANK repeats (*Figure 4C*). Perturbations of these hydrogen bonding interactions (e.g. the Asn834Lys substitution in AI-c or the Asn595Ala mutation in one of the ANK repeats finger loop) led to a few fold decreases in their bindings (*Figure 4D*). Hydrophobic interactions also contribute to the AI-c autoinhibition. Met839 is buried in the hydrophobic pocket formed by Leu668 and Leu701; Leu843 interacts with sidechains of Leu597, Tyr630, and Ile635

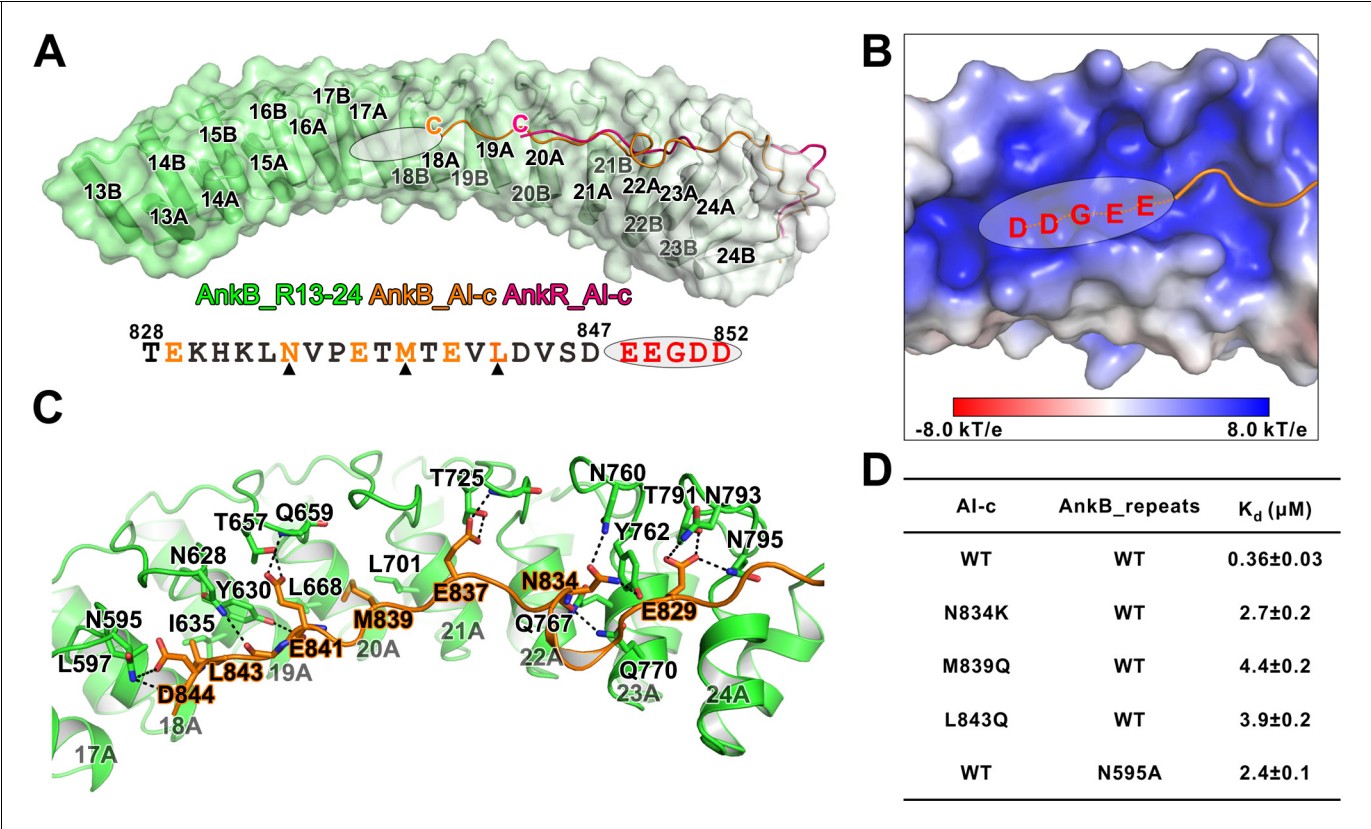

**Figure 4.** Detailed interaction between AnkB_AI-c and AnkB_MBD. (**A**) Comparison of the structures of the AnkB_R13-24/AnkB_AI-c complex (colored in orange) and AnkR_C12 (colored in hot pink). The ANK repeats of AnkB are shown in cylinder and transparent surface and ANK repeats of AnkR are omitted for clarity. The folded back inhibitory sequence immediately following the ANK repeats in both structures are shown using the line model. The position of the stretch of negatively charged residues, which are not defined in the crystal structure, is indicated with a white oval. The amino acid sequence of AnkB_AI-c is also shown below the structure. Residues critical for the binding are shown in orange with those verified by mutagenesis highlighted by black triangles. (**B**) The charge potential surface of the inner groove formed by R16-17, calculated by the APBS module embedded in PyMOL and contoured at ±8 kT/e. (**C**) Detailed interactions between AnkB_AI-c and AnkB_MBD. (**D**) ITC derived dissociation constants showing mutations of critical residues weaken or abolish the binding between AnkB_AI-c and AnkB_MBD.

DOI: https://doi.org/10.7554/eLife.29150.008

(*Figure 4C*). Individually mutating these two hydrophobic residues to polar Gln weakened the binding by several folds (*Figure 4D*).

An ideal result would be to obtain structures of MBD of ankyrins binding to two or even all three of these autoinhibitory segments. We have extensively tried such experiments, by fusing the inhibitory fragments combining with the different strategies used in the study. Unfortunately, these extensive efforts have not resulted fruition likely due to the conformational dynamics of the elongated protein complexes.

## The autoinhibition segments inhibit AnkB's plasma membrane localization

In epithelial cells, AnkG clusters on plasma membranes whereas AnkB largely localizes to intracellular compartments (*Figure 5A and B* and *He et al., 2013*). Autoinhibition of the MBD by the linker region can modulate AnkB's membrane *vs* cytosol distributions in epithelial cells (*He et al., 2013*). We made use of this assay as a functional readout to verify the autoinhibited structures determined in this study and to provide a preliminary glance at the role of AnkB's autoinhibition. Consistent with the previous report (*He et al., 2013*), WT AnkB mainly localizes in the cytosol, whereas WT AnkG mainly associates with the plasma membranes in polarized MDCK cells (*Figure 5A and B*). We constructed two mutants in the linker region of AnkB, one weakens the AI-b's binding to MBD (the

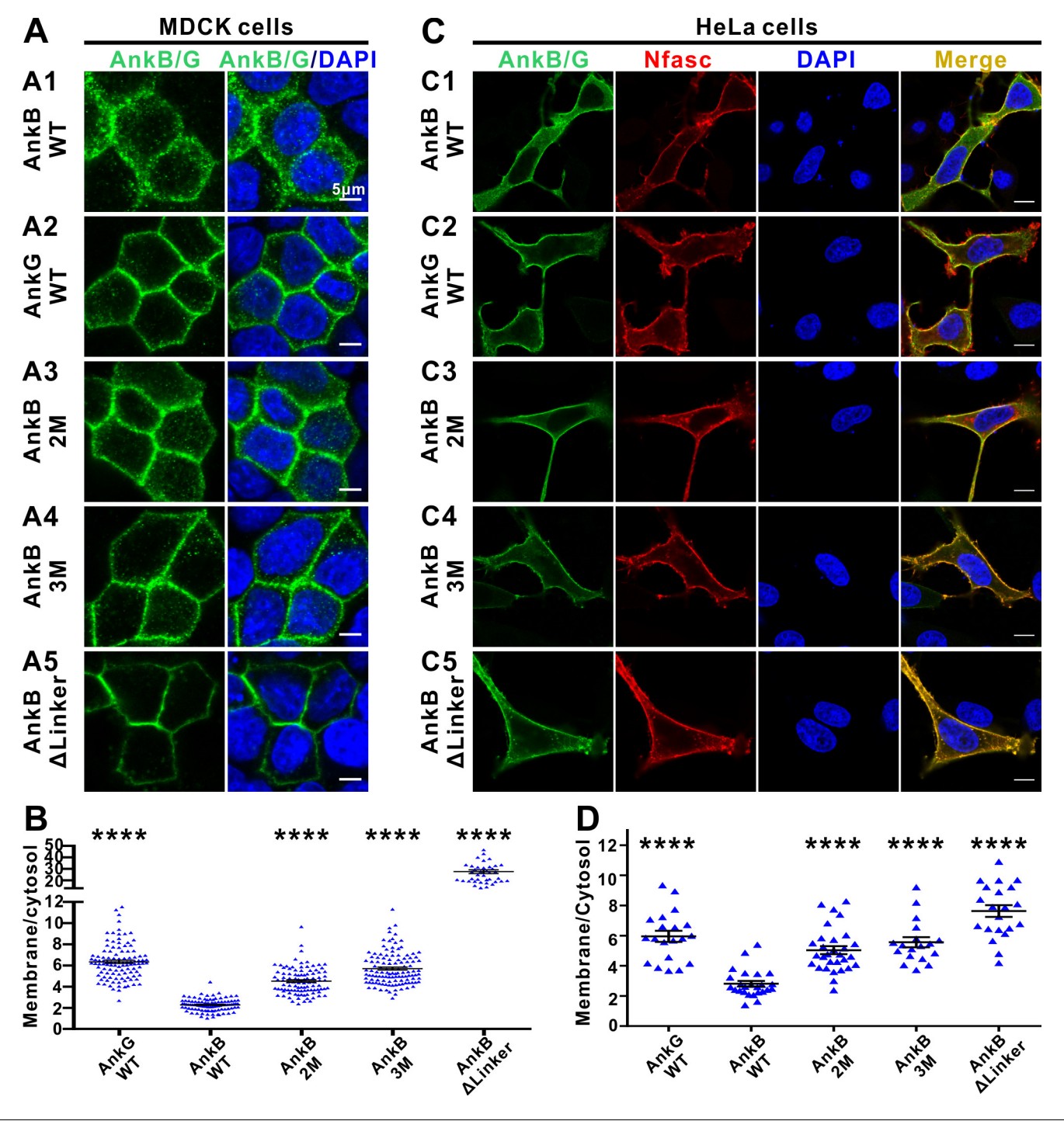

**Figure 5.** The autoinhibitory segments regulate subcellular localization of AnkB in MDCK cells and NF186-dependent membrane recruitment of AnkB in HeLa cells. (**A**) Representative fluorescent images of transiently expressed GFP-tagged AnkG, AnkB or its linker mutants in polarized MDCK cells with nuclei stained with DAPI (blue): A1, WT AnkB; A2, WT AnkG; A3, AnkB_2M; A4, AnkB_3M; A5, AnkB ΔLinker. (**B**) Quantification of the immunofluorescence intensity ratio of plasma membrane *vs* cytosolic GFP signals. Data are presented as means ± SEM from>100 cells (except for AnkB ΔLinker with 41 cells due to its very clear membrane localizations) and analyzed using one way ANOVA followed by Dunnett's multiple comparisons test to WT AnkB, ****p<0.0001. (**C**) Representative fluorescent images of HeLa cells transiently co-expressing HA-tagged NF186 (red) and GFP-tagged AnkG, AnkB or its linker mutants (green), with nuclei stained with DAPI (blue): C1, WT AnkB; C2, WT AnkG; C3, AnkB_2M; C4, AnkB_3M; C5, AnkB ΔLinker. (**D**) Quantification of the immunofluorescence intensity ratio of plasma membrane *vs* cytosolic GFP signals (representing AnkB/G level). Data are presented as means ± SEM from~20 cells and analyzed using one way ANOVA followed by Dunnett's multiple comparisons test to WT AnkB,

*Figure 5 continued on next page*

*Figure 5 continued*

****p<0.0001. The protein expression levels of all constructs in both cell lines are comparable as indicated by the quantified total fluorescence intensities of each groups in this experiment.
DOI: https://doi.org/10.7554/eLife.29150.009

Met884Ala, Tyr886Ala double point mutations, denoted as AnkB 2M) and the other weakens both AI-b and AI-c's bindings to MBD (the Met884Ala, Tyr886Ala, Asn834Lys triple mutations, denoted as AnkB 3M), and assayed their membrane *vs* cytosol distributions in polarized MDCK cells. We observed that the AnkB 2M and 3M mutants show a higher ratio of plasma membrane localization (*Figure 5A and B*), consistent with releases of the autoinhibition induced by the two mutations. Finally, we deleted essentially the entire linker region encompassing the complete AI-b and AI-c segments (AnkB ΔLinker, with aa 828–943 deleted), and found that this deletion mutant is near completely membrane localized (*Figure 5A and B*), suggesting that both AI-b and AI-c can regulate AnkB's membrane localization presumably by modulating its MBD's binding to membrane-anchored target(s). We have also performed NF186-mediated plasma membrane recruitments of AnkB, and compared the impacts of the 2M and 3M mutations on AnkB's membrane localizations in HeLa cells which lack endogenous NF186 expression. Co-expression of NF186 partially recruited WT AnkB to the plasma membranes (*Figure 5C*). The 2M, 3M, and ΔLinker of AnkB mutants displayed sequentially increasing amount of NF186-mediated membrane recruitments in this assay system (*Figure 5C and D*). Similar as observed in MDCK cells, WT AnkG is better recruited to plasma membranes by NF186 than WT AnkB is (*Figure 5C and D*).

## AnkG and AnkB share a similar autoinhibition mechanism

It is not known whether AnkG also contains auto-regulatory segments. Amino acid sequence alignment analysis reveals that residues corresponding to the AnkB AI-c are essentially the same in AnkG, indicating that AnkG also contains an autoinhibitory AI-c segment (*Figure 6A*). Curiously, the N-terminal half of AnkB AI-b can be nicely aligned with a fragment in the AnkG linker following AI-c, but residues in the C-terminal half of AnkB AI-b that are critical for its binding to MBD (e.g. Met884 and Tyr886) are missing in AnkG (*Figure 6A*). We hypothesized that AnkG may contain a different auto-inhibitory AI-b segment from that of AnkB. Using a similar binding assay developed to discover AI-b in AnkB, we found that elongating the linker from residue 855 to 903 in the extended AnkG MBD (compare aa 38–855 *vs* aa 38–903) weakened the bindings of Nav1.2, E-cadherin and NF186 to AnkG by 6 ~ 74 folds (*Figure 6B*), indicating that AnkG linker indeed contains an AI-b segment. We also detected direct interaction between AnkG MBD-AI-c protein (aa 38–855, containing the AI-c segment) and an AnkG's long linker (aa 837–920) or a shorter version of the linker (aa 861–920, lacking AI-c) (*Figure 6C and D*). Binding affinity-guided mapping revealed that residues 875 to 903 encompass the complete AI-b of AnkG (*Figure 6E*). It is noted that AnkG AI-b contains a stretch of highly charged residues in its N-terminal half, and these residues are not found in AnkB, revealing a different binding mechanism of AI-b to MBD in AnkB and G. We found that increasing salt concentrations in the binding buffer significantly weakened the interaction between AnkG_AI-b and AnkG_MBD, supporting the charge-charge interactions between AI-b and AnkG_MBD (data not shown). We substituted three charged residues with neutral or charge reverse residues (Asp887Ala, Asp889Ala and Lys890Glu, indicated with black triangles in *Figure 6A*) from the AnkG linker (861-920) and found that the mutant completely lost its ability to bind AnkG_MBD (*Figure 6F*), further supporting the critical roles of the charged residues in AnkG AI-b in its autoinhibition.

We attempted to map the AI-b's binding site on AnkG's ANK repeats by a similar repeats truncation approach used for AnkB (*Figure 3B*), but this effort failed due to poor sample behaviors of the AnkG_MBD truncation mutants. Alternatively, we generated the 'site-1, -2, and -3' target binding mutants of AnkG_MBD based on our earlier work (*Wang et al., 2014*), and tested each of these mutants in binding to the AnkG_AI-b fragment. We found that mutations in "site-1' (Phe141Gln, Phe164Gln) or in 'site-2' (Ile277Gln, Leu310Gln) of AnkG_MBD has little effect on AI-b's binding, whereas mutations in 'site-3' (Leu376Gln, Phe409Gln, Leu442Gln) completely abolished the interaction between AI-b and ANK repeats (*Figure 6G*). This result indicates AnkG_AI-b also chiefly binds to the 'site-3' (repeats R11-14) of AnkG_MBD. The above result, together with the findings that

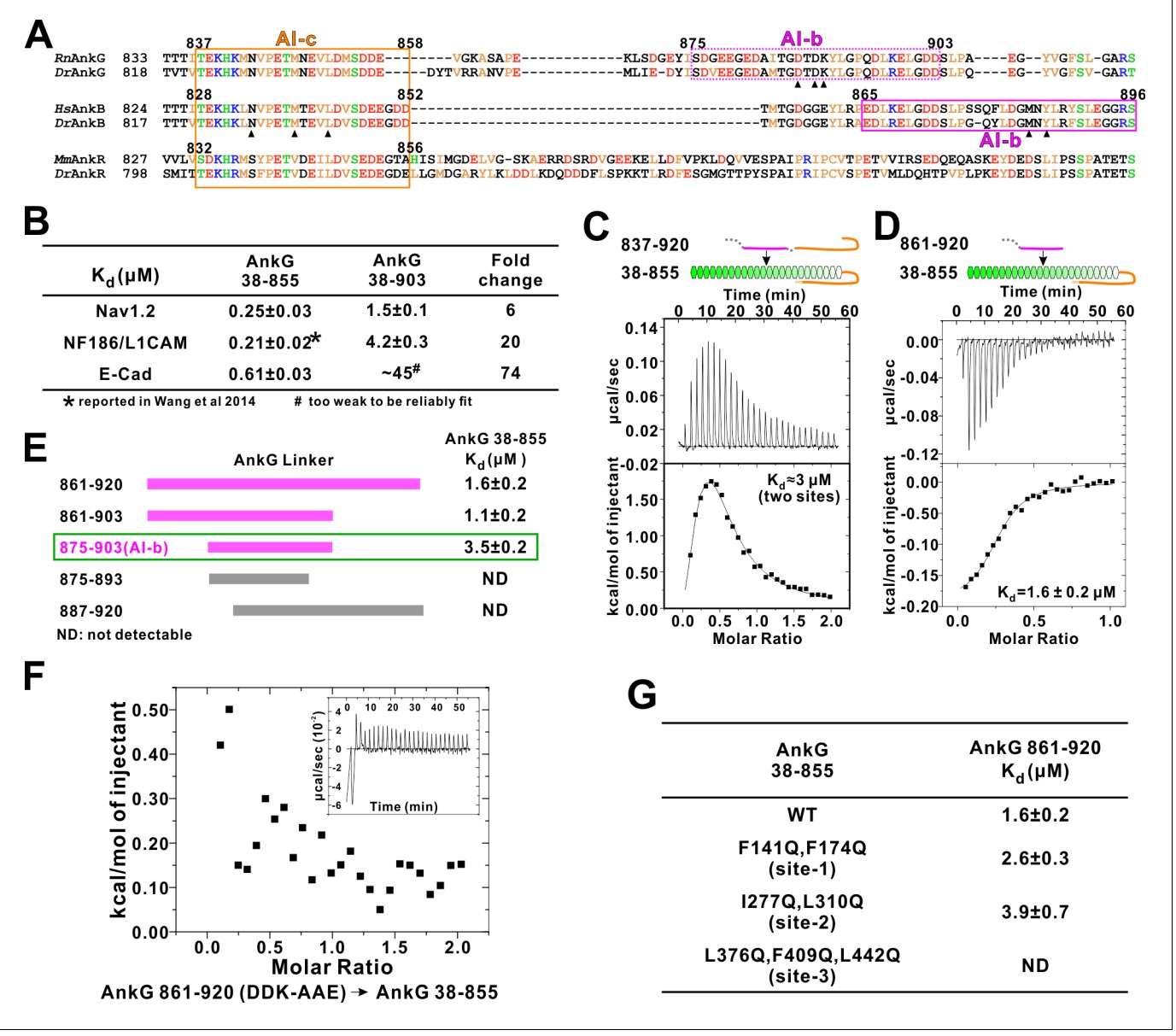

**Figure 6.** Autoinhibition of AnkG MBD by the linker region. (A) Amino acid sequence alignment of the linker regions of AnkG, AnkB, and AnkR. The boundaries of AI-c and AI-b of AnkB and G are indicated with orange and magenta boxes. Critical residues that have been verified by mutagenesis are highlighted with black triangles. (B) ITC-derived binding affinities showing that including longer linker region to the AnkG_MBD weakened the binding towards its binding targets Nav1.2, NF186/L1CAM, and E-cadherin. (C and D) ITC profiles showing direct binding of the linker of AnkG to AnkG_MBD: C, AnkG 837–920, a linker containing both AI-b and AI-c binding to AnkG_MBD; D, AnkG 861–920, a linker containing only the AI-b, binding to AnkG_MBD. (E) ITC-based mapping of the AnkG_AI-b region. The minimal region of AI-b is indicated with a green box. (F) ITC result showing that mutating the charged residues 'DDK' to Ala or charge reversed residue disrupted the binding between AnkG 861–920 to AnkG 38–855. (G) ITC-derived dissociation constants of AnkG 861–920 to AnkG WT and its 'sites-1,2,3' mutants showing that AnkG 861–920 predominantly binds to the 'site-3' of AnkG_MBD.

DOI: https://doi.org/10.7554/eLife.29150.010

AnkG also contains the inhibitory AI-a segment (*Figure 1B*) and AI-c segment (*Figure 6A*), collectively demonstrated that AnkG shares a very similar, three segment-mediated MBD autoinhibitory mechanism as AnkB does, although the AI-b segments in the two ankyrins are different.

## Discussion

### Features of the AnkB and G autoinhibition

Ankyrins are master scaffold proteins assembling very diverse signaling microdomains beneath membrane bilayers. This is achieved by their MBD-mediated bindings to numerous trans-membrane proteins and SBD-mediated anchoring of the protein complex to spectrin-based cytoskeletal meshwork. Formation of a highly elongated and malleable target binding groove by the 24 ANK repeats with multiple semi-independent target binding sites provides a mechanistic explanation to how ankyrin MBD, via combinatorial usages of its target binding sites, can bind to many distinct membrane targets with high specificity (*Figure 7A*; and *Wang et al., 2014*). A key unanswered question is how MBD-mediated recognitions of membrane targets of ankyrins are regulated. Autoinhibition by C-terminal segment(s) to the ANK repeats has been suggested to be one of such regulation mechanisms (*Abdi et al., 2006*).

In this study, we characterized the autoinhibition of AnkB and G in detail and discovered that the entire target binding grooves of both AnkB and G MBDs could be inhibited by multiple discrete and intrinsically disordered peptide fragments located in the MBD/SBD linker region and in the tail region right after DD (*Figure 7*). Except for the AI-c segment located immediately C-terminal to the ANK repeats, the inhibitory sequences of AI-a and AI-b segments between AnkB and AnkG share limited homology. This is perhaps correlated with the differential potential target binding regulations of the two MBDs. However, the amino acid sequences of all three identified autoinhibitory segments are highly conserved throughout the evolution both for AnkB and for AnkG, implying that the autoinhibition mechanisms discovered here are evolutionary conserved features for the two ankyrins, respectively. Given that the inhibitory segments of both AnkB and G are intrinsically disordered and disordered sequences tend to evolve more rapidly in eukaryotic genomes (*Dyson and Wright, 2005*), such high amino acid sequence conservation suggests a strong function-mediated selection against mutational drifts of these inhibitory segments. Expressions of both AnkB and G can be extensively diverse due to their alternative splicing, and therefore the expressed proteins can vary in their sizes dramatically (*Cunha and Mohler, 2009*). However, regardless of the extensive alternative splicing, the autoinhibitory segments AI-b and AI-c are always retained in all documented AnkB or AnkG variants according to the *Ensembl* database (http://www.ensembl.org), suggesting that the autoinhibitory mechanisms characterized here for AI-b and c are common to all isoforms of AnkB and G. Interestingly, the giant AnkG isoforms (270 kDa/480 kDa AnkG) lose the AI-a segment due to alternative splicing; whereas giant AnkB (440 kDa AnkB) keeps this AI-a segment, suggesting the differential regulation of these giant ankyrin isoforms in nervous systems. Our data is also consistent with the findings from super-resolution images (*Leterrier et al., 2015*). In AIS, 270 kDa/480 kDa AnkG's SBD exhibits a periodic pattern, while their C-terminus part is not periodically arranged and is found ~32 nm radially deeper than the SBD in the axoplasm. AIS AnkG (270 kDa/480 kDa isoforms) lacks AI-a, so the major parts of CT of 270 kDa/480 kDa AnkG are not sequestered by MBD even if AI-b and c are in contact with MBD. Furthermore, the 270 kDa/480 kDa AnkG at AIS/Nodes of Ranvier is likely to adopt an open conformation. As such, the signals of antibodies labeling the C-terminus part of AnkG should not appear near the membrane but is deeper into the axoplasm.

Since each target protein normally occupies a few of the total five proposed binding sites along the entire 24 ANK repeats (*Figure 7A*), the use of multiple semi-independent autoinhibitory segments lining the entire target binding groove of MBD can provide a mechanism for selected release of a few target binding sites (e.g. sites-1 and 3 for sodium channels) while keeping other sites closed (e.g. site-5), thus allowing selected engagements of MBD to certain target proteins at any given membrane microdomains. The combinatorial usage of multiple binding sites by target proteins appears to be a rather common strategy for many long repeat-containing proteins (*Chook and Blobel, 2001*; *Conti et al., 1998*; *Graham et al., 2000*; *Huber and Weis, 2001*; *Kobe, 1999*; *Xu et al., 2010*; *Zhu et al., 2011*). Autoinhibition of an elongated repeats domain by multiple intrinsically disordered segments from the same protein are rather uncommon, and only a few cases have been reported. Kap60p, a karyopherin family protein, adopts a somewhat similar autoinhibition mode as ankyrins do. In this case, two NLS recognition sites on the armadillo repeats were shown to be autoinhibited by two consecutive segments from Kap60p's N-terminal unstructured region (*Matsuura and Stewart, 2004*). In another case, two NLS binding sites of importin-α are inhibited by the same fragment from its N-terminal unstructured region at a 1:2 stoichiometry instead of

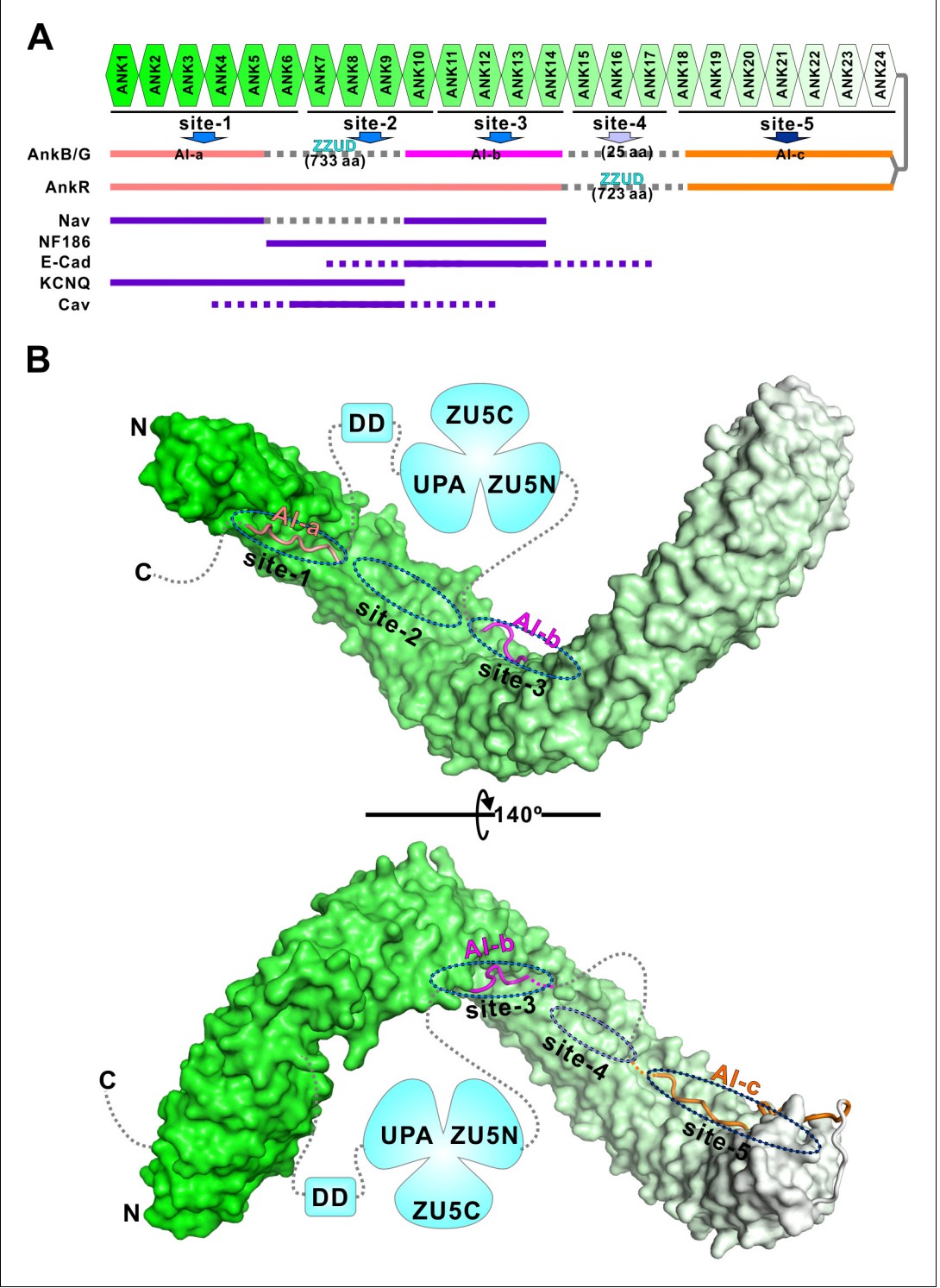

**Figure 7.** Combinatorial autoinhibitions of ankyrins and their potential regulations. (**A**) Schematic diagram showing the combinatorial autoinhibition mechanisms of the three Ankyrin family members. 'Sites-1,2,3' are drawn according to our earlier study (*Wang et al., 2014*). R15-17 is assigned as 'site-4' based on the biochemical and structural analysis in this work and the concept that three to five ANK repeats can form a stable structural unit for target recognition, although no targets have been characterized to date. R18-24 is assigned as 'site-5' since it is autoinhibited by AI-c. The MBD binding sequences of the autoinhibitory segments or reported targets are shown in solid lines at corresponding positions of the bind sites. (**B**) Structural model of the autoinhibited full-length AnkB constructed with the three structures solved in the current study. The MBD is shown in surface

*Figure 7 continued on next page*

*Figure 7 continued*

representation. The three autoinhibitory segments are shown with the worm model. The unstructured loops are shown in grey dashed lines.

DOI: https://doi.org/10.7554/eLife.29150.011

multiple segments with different sequences as found in ankyrins and Kap60p (*Catimel et al., 2001*). Other autoinhibition mechanisms found in long repeat-containing proteins include the HEAT repeats of karyopherin-β2 occupied by an internal long loop from the middle region of the HEAT repeats (*Chook and Blobel, 1999*; *Lee et al., 2006a*), the HEAT repeats of exportin chromosome region maintenance 1 (CRM1) autoinhibited by a C-terminal α helix immediately following the HEAT repeats (*Saito and Matsuura, 2013*), and the armadillo repeats of Diaphanous-related formins (Drfs) autoinhibited by a short α helix from its distal C-terminal region (*Lammers et al., 2005*). It will be interesting to investigate whether the autoinhibition by multiple discrete segments as observed in ankyrins here might also be used by other long repeat-containing proteins capable of binding to diverse targets in the future.

## Functional implications and potential regulations of ankyrin autoinhibition

Ankyrin MBDs are versatile membrane target binders and many of these ankyrin binding membrane proteins are located in specific membrane microdomains (e.g. ion-channels in the axon initial segments of neurons as well as specifically patterned membrane domains in cardiomyocytes) playing vital physiological roles. Defects of interactions between ankyrins and their membrane targets are frequently linked to human diseases including hereditary spherocytosis, cardiac arrhythmia, and several types of psychiatric disorders (*Eber et al., 1996*; *Mohler et al., 2004b*; *Van Camp et al., 1996*). In view of the vital physiological roles of the interactions between ankyrins and these membrane targets, a priori assumption is that such ankyrin-mediated membrane target bindings must be regulated. For example, the membrane target binding groove of MBDs is likely to be closed, via the autoinhibition mechanism elucidated in this study, during biogenesis and trafficking processes of ankyrins. Upon reaching each specific membrane domain, certain mechanisms are likely existing to release a few selected target binding sites for specific assembly of ankyrin/membrane target complexes. The existence of multiple semi-independent inhibitory sequences in the tail of ankyrins suggests that these sequence may be regulated in a combinatorial fashion so that ankyrins can scaffold many different target proteins in different cellular settings and in different tissues.

The positioning of the spectrin binding ZZUD domains with respect to the auto-inhibitory sites-a and b suggests a possible spectrin/SBD interaction-induced release of ankyrin autoinhibition. The spectrin binding ZZUD tandem are flanked by the AI-a and AI-b segments both in 220 kDa AnkB and 190 kDa AnkG, and the connection sequences between the ZZUD tandem and the two inhibitory segments are not very long (*Figures 1A* and *7B*). Once ankyrins and spectrins meet at specific membrane microdomains, both the bulky ankyrin SBD/spectrin complex as well as the perceived 'pulling force' exerted by the spectrin-bound SBD may dislodge the two inhibitory segments, AI-a and AI-b, flanking the ZZUD tandem and thus lead to the release of the autoinhibition. Accompanied by the spectrin binding-induced release of the AI-a and AI-b segments, the target binding 'sites-1,2,3' are opened and the corresponding membrane proteins can bind to the ankyrin MBDs. It should be noted that such coordinated membrane microdomain targeting and spectrin/actin network binding-induced release of ankyrin MBD autoinhibition model, if it is indeed employed in cells, is likely used together with other regulatory mechanisms in regulating the conformational opening of ankyrins. Ankyrins at AIS/nodes of Ranvier are chiefly 270 kDa/480 kDa AnkG isoforms. Interestingly, these two larger AnkG isoforms do not contain the 'AI-a' sequence in their CTs due to alternative splicing. Thus, the 270 kDa/480 kDa AnkG seem to have different autoinhibition mechanisms with respect to other ankyrins (e.g. 190 kDa AnkG and 220 kDa AnkB). The lack of the 'AI-a' site in 270 kDa/480 kDa AnkG means that these two AnkG isoforms can still bind to Nav channels or NF186 (See *Figure 2B*, AnkB 28–965; and also *Hedstrom et al., 2008*), albeit with much lower affinities (i.e. a form of semi-open conformation). Spectrin binding to the AnkG SBD may promote further opening of ANK repeats and thus leading to full engagements of the membrane targets of AnkG.

Such mutual reinforcement of AnkG-mediated membrane microdomain assembly model is consistent with the findings showing that membrane targets and βIV spectrin mutually stabilize each other through AnkG at AIS or nodes of Ranvier (*Yang et al., 2004*; *Patzke et al., 2016*). Additionally, post-translational modifications (e.g. phosphorylation by different protein kinases) can provide another level of differential activations of ankyrins (see below).

Although AnkB and AnkG share the similar overall autoinhibition modes, the detailed amino acid sequences in their autoinhibitory segments are quite different (*Figures 6A* and *7A*). In parallel with this finding, AnkB and AnkG are often targeted to different subdomains in the same tissues including neurons and cardiomyocytes (*Galiano et al., 2012*; *Lowe et al., 2008*; *Mohler et al., 2005*). It is possible that the unique sequences of their inhibitory segments provide means for AnkB and G to be differentially regulated. One such possible mean is via post translational modifications such as phosphorylations (*Lu et al., 1985*; *Cianci et al., 1988*). We surveyed the PhosphoSitePlus database (http://www.phosphosite.org/) and found that Ser855 in AI-c, Thr888, Tyr891 in AI-b of AnkG and Thr838, Ser846 in AI-c, Tyr889, Ser890 in AI-b of AnkB can be phosphorylated. These residues are positioned within the autoinhibitory segments. It is possible that phosphorylation on some of these residues may lead to differential releases of the MBD autoinhibition of AnkB or AnkG. Again, the disordered nature of the autoinhibitory segments of ankyrins is favorable for their accesses to different protein kinases. It is further noted that different ankyrin-enriched membrane microdomains (for example AIS) are often selectively enriched with certain protein kinases (*Bréchet et al., 2008*; *Hund et al., 2010*; *Tapia et al., 2013*). Post translational modifications on the intrinsically disordered inhibitory segments of ankyrins as a mean of their target recognition regulations may be a fertile ground to explore in the future.

In summary, we systemically studied the autoinhibition mechanisms of the 24 ANK repeats of AnkB and G by their own respective MBD/SBD linker and proximal tail segments. We biochemically characterized these intramolecular interactions, solved the representative complex structures and quantitatively evaluated the inhibitory effects of these autoinhibitory segments on MBD's bindings to different physiological membrane targets. We demonstrated that the combinatorial uses of multiple semi-independent, intrinsically disordered autoinhibitory segments can provide graded regulations of targets binding by ankyrin MBDs. Our findings on the autoinhibition as well as target bindings of the 24 ANK repeats of ankyrins may also shed light on target recognitions and regulations by other long repeat-containing scaffold proteins that are quite abundant in the mammalian proteomes.

## Materials and methods

### Constructs, protein expression and purification

The coding sequences of the AnkR constructs were PCR amplified from a mouse muscle cDNA library. The coding sequences of AnkB and AnkG constructs were PCR amplified from the full-length human 220 kDa AnkB template or the full-length rat 270 kDa AnkG template respectively (both templates as well as the HA tagged full length NF186 construct are generous gifts from Dr. Vann Bennett). E-cadherin (aa 734–884), NF186 (aa 1187–1214), L1CAM (aa 1206–1233) and Nav1.2 (aa 1035–1129) coding sequences were PCR amplified from mouse brain or muscle cDNA libraries. All of the constructs that used for protein expression were cloned onto a home-modified pET32a vector. All truncation mutations of ANK repeats constructs were made with the same strategy as described in our previous study (*Wang et al., 2014*). The fusion constructs of AnkR/B_CT Chimera/AnkB_repeats_R1-20 and AnkB_AI-b/AnkB_R8-M14 were made by standard two-step PCR with a coding sequence of 'GSLVPRGSGS' as the flexible linker (M14 means replacing the αB of R14 with a capping sequence corresponding to the αB of R24 for protein stabilization as we used earlier). The same strategy was used in making other fusion constructs described in this study. All point mutations were created using the Quick Change site-directed mutagenesis kit and confirmed by DNA sequencing. Protein expression and purification protocols are the same as previously described (*Wang et al., 2012*; *2014*). Recombinant proteins were expressed in BL21 (DE3) *Escherichia coli* cells with induction of 0.25 mM IPTG at 16°C. The N-terminal Trx-his$_6$-tagged proteins were purified using Ni$^{2+}$-NTA agarose affinity column followed by size-exclusion chromatography (Superdex 200 column

from GE Healthcare, Little Chalfont, UK) in the final buffer containing 50 mM Tris-HCl, 1 mM DTT, and 1 mM EDTA, pH 7.8 with either 100 mM NaCl or 500 mM NaCl as required.

For simplicity, we use human 220 kDa AnkB (NM_020977.3), rat AnkG (MBD-UPA: NM_001033984.1 and DD-CT: NM_031805.1) and mouse AnkR (NM_001110783.3) for the amino acid numbering throughout the manuscript. The various constructs of ankyrins used in this study are listed in *Table 2*.

## Isothermal titration calorimetry assay and fluorescence assay

Isothermal titration calorimetry (ITC) assays and fluorescence assays were carried out with the same protocol as described earlier (*Wang et al., 2014*). Briefly, isothermal titration calorimetry assays were performed on a VP-ITC MicroCal calorimeter (MicroCal, Northampton, MA) at 25°C and data were analyzed and fitted using the program Origin7.0 (Microcal). Fluorescence-based binding assays were performed on a PerkinElmer LS-55 fluorimeter equipped with an automated polarizer at 25°C. The Kd values were obtained by fitting the titration curves with the classical one-site binding model.

## Crystallography

All crystals were obtained by hanging drop or sitting drop vapor diffusion methods at 16°C. Crystals of AnkR/B_CT Chimera/AnkB_repeats_R1-20 were grown in solution containing 4 M ammonium acetate and 0.1 M Bis-Tris propane (pH 7.0). Crystals of AnkB_AI-b/AnkB_R8-M14 were grown in solution containing 0.1 M HEPES (pH 7.0), 1 M ammonium sulfate and 0.5% w/v PEG 8,000. Crystals of AnkB_AI-c/AnkB_R13-24 were grown in solution containing 0.2 M $CaCl_2$, 0.1 M HEPES (pH 7.5) and 28% v/v PEG 400. Crystals were soaked in crystallization solution containing additional 20% glycerol for cryoprotection. All datasets were collected at the Shanghai Synchrotron Radiation Facility at 100 K. Data were processed and scaled using HKL2000 (*Otwinowski and Minor, 1997*).

Structures were solved by molecular replacement using PHASER (*McCoy et al., 2007*) with fragments of the entire 24 ANK repeats (PDB: 4RLV) as the searching models. Peptides were manually built according to $F_o$-$F_c$ difference maps in COOT (*Emsley et al., 2010*). Further manual model adjustment and refinement were completed iteratively using COOT (*Emsley et al., 2010*) and PHENIX (*Adams et al., 2010*). The final models were validated by MolProbity (*Chen et al., 2010*) and statistics are summarized in *Table 1*. All structure figures were prepared by PyMOL (http://www.pymol.org). The coordinates of the structures reported in this work have been deposited to PDB under the accession codes of 5Y4D, 5Y4E and 5Y4F for the RB-Chimera/AnkB_R1-20, AI-b/AnkB_R8-M14 and AI-c/AnkB_R13-24 structures, respectively.

## Cell culture, transfection and immunostaining

MDCK cells were seeded on 35 mm dishes with 10 mm diameter uncoated glass bottom (MatTek, Ashland, MA) and grown in 10% FBS supplemented DMEM at 37°C incubator with 5% $CO_2$. After around 20 hr, when the confluency reached 30 ~ 40%, cells were transfected with 300 ng plasmids using Lipofectamine 2000 transfection reagent (Invitrogen, Carsbad, CA) following the protocol suggested by the manufacturer. After transfection, MDCK cells were grown in 10% FBS supplemented DMEM until they were fully polarized. HeLa cells were cultured in the same media and culture condition as used for MDCK cells. HeLa cells were co-transfected with 400 ng HA tagged NF186 plasmids and 500 ng ankyrin constructs using Viafect transfection reagent (Promega, Madison, WI) when the confluency reached 20 ~ 30%. Then cells were grown in 10% FBS supplemented DMEM for 24 hr and fixed. HeLa (RRID: CVCL_0030) and MDCK (RRID: CVCL_0422) cells were originated from ATCC. These cells were not individually authenticated and not found to be on the list of commonly misidentified cell lines (International Cell Line Authentication Committee). Cells were tested negative for mycoplasma contamination by cytoplasmic DAPI staining.

The MDCK or HeLa cells were fixed with 4% paraformaldehyde at room temperature for 15 min, and permeabilized with 0.2% Triton X-100 at room temperature for 15 min followed by a 60 min blocking in PBS buffer containing 5% bovine serum albumin. For immunostaining, cells were then incubated with primary antibodies (goat anti GFP, ab6658, Lot: GR206330-6, RRID: AB_305631; Rabbit anti HA, Sigma, H6908, RRID: AB_260070) at 4°C overnight. The next day, cells were washed with PBS buffer three times and then incubated with fluorescence-conjugated secondary antibodies

(Alexa Fluor 488 or 594) at room temperature for 2 hr, followed by incubating with 500 nM DAPI for 5 min to stain nucleus. Then cells were washed with PBS before imaging.

## Microscopy and data analysis

All the cell culture images were captured by a Zeiss LSM 880 laser-scanning confocal microscope. The MDCK cell and HeLa cell images were captured using a 40 × 1.4 oil objective with pinhole setting to 1 Airy unit. Fluorescence intensity were analyzed with ImageJ software (https://imagej.nih.gov/ij/) and statistically analyzed with GraphPad Prism five using one-way ANOVA followed by Dunnett's multiple comparisons test.

## Acknowledgements

We thank the Shanghai Synchrotron Radiation Facility (SSRF) BL19U1 and BL17U1 for X-ray beam time. We also thank Dr. Vann Bennett for valuable discussions and experimental materials from his laboratory at Duke University. This work was supported by grants from RGC of Hong Kong (663812, 664113, 16103614, and AoE-M09-12), and a 973 program grant (2014CB910204) and a National Key R and D Program (2016YFA0501900) from the Minister of Science and Technology of China to MZ. MZ is a Kerry Holdings Professor in Science and a Senior Fellow of IAS at HKUST.

## Additional information

### Competing interests

Mingjie Zhang: Reviewing editor, *eLife*. The other authors declare that no competing interests exist.

### Funding

| Funder | Grant reference number | Author |
| --- | --- | --- |
| Research Grants Council, University Grants Committee | 663812 | Mingjie Zhang |
| Ministry of Science and Technology of the People's Republic of China | 2014CB910204 | Mingjie Zhang |
| Research Grants Council, University Grants Committee | 664113 | Mingjie Zhang |
| Research Grants Council, University Grants Committee | 16103614 | Mingjie Zhang |
| Research Grants Council, University Grants Committee | AoE-M09-12 | Mingjie Zhang |
| Ministry of Science and Technology of the People's Republic of China | 2016YFA0501903 | Mingjie Zhang |

The funders had no role in study design, data collection and interpretation, or the decision to submit the work for publication.

### Author contributions

Keyu Chen, Jianchao Li, Data curation, Formal analysis, Investigation, Writing—original draft, Writing—review and editing; Chao Wang, Zhiyi Wei, Data curation, Formal analysis, Investigation, Writing—review and editing; Mingjie Zhang, Conceptualization, Supervision, Writing—original draft, Writing—review and editing

### Author ORCIDs

Keyu Chen (ID) http://orcid.org/0000-0003-0321-0604
Jianchao Li (ID) http://orcid.org/0000-0002-8921-1626

Zhiyi Wei ⓘ http://orcid.org/0000-0002-4446-6502
Mingjie Zhang ⓘ http://orcid.org/0000-0001-9404-0190

**Decision letter and Author response**
Decision letter https://doi.org/10.7554/eLife.29150.013
Author response https://doi.org/10.7554/eLife.29150.014

## Additional files

**Supplementary files**
• Transparent reporting form
DOI: https://doi.org/10.7554/eLife.29150.012

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
