## [Decision Letter]

Thank you for submitting your article "Autoinhibition of ankyrin-B/G membrane target bindings by intrinsically disordered segments from the tail regions" for consideration by *eLife*. Your article has been reviewed favorably by three peer reviewers, and the evaluation has been overseen by a Reviewing Editor and John Kuriyan as the Senior Editor. The following individuals involved in review of your submission have agreed to reveal their identity: Yuh Min Chook (Reviewer #1); Peter Michaely (Reviewer #2).

The reviewers have discussed the reviews with one another and the Reviewing Editor has drafted this decision to help you prepare a revised submission. Note that the essential revisions listed below do not require additional experiments, but please discuss your response to each of the points thoroughly, and revise the manuscript accordingly.

Summary:

The manuscript from the Zhang group revealed how ankyrin proteins are autoinhibited by three disordered segments – two from the linker region and one from the C-terminal disordered region of the proteins. They used a combination of quantitative mapping by ITC and X-ray crystallography to identify these three segments in all three ankyrin isoforms AnkB, AnkG and AnkR. The finding is important as this work provides the first mechanistic explanation of how binding of the ankyrins to different membrane targets that occupy different sites on the ankyrin MBD may be regulated differentially as the three autoinhibitory segments each occupy different sites along the ankyrin repeat solenoid.

Essential revisions:

1) This point concerns limitations of the model proposed by the authors. The authors propose a mechanism whereby binding of ankyrins to spectrins releases the AI domains allowing them to interact with their membrane binding targets. This model is not consistent with the fact that AnkG recruits bIV spectrin to the AIS, and that NF186 recruits ankG to nodes of Ranvier. Clustered ankG recruits spectrins to the AIS and nodes (see Yang et al., JCB), and clustering of ankG at the AIS does not require βIV spectrin (see Hedstrom et al., JCB) to cluster Na^+^ channels or NF186. While the model is plausible, the authors should discuss the limitations of the model. Moreover, the authors hypothesize that association of ankyrin with spectrin activates MBD associations with client membrane proteins. This hypothesis is attractive, but on its surface, seems to be in conflict with published data showing that full-length ankyrins purified from tissue can associate with high affinity to at least many of ankyrins' membrane protein clients in the absence of spectrin (e.g., PMID: 2452986, 3039371, 379653). The authors also hypothesize that phosphorylation events in the CTD may activate specific client associations by displacing CTD sequences blocking aspects of the MBD. This too is an attractive hypothesis and is one that has previously been suggested (e.g., PMID 2933395, 2970468). It may be that ankyrins are synthesized in a "closed" conformation, and upon interaction with spectrin, adopt a semi-open conformation that makes C-terminal sequences targets for phosphorylation. These phosphorylation events may then activate membrane protein binding activity. Ankyrins purified from tissues may have existent CTD phosphorylations that allow interaction with membrane proteins. As also discussed by the authors, the CTD is subject to extensive alternative splicing resulting in different ankyrins with different regulatory sequences. mRNA splicing events that eliminate specific blocking sequences may promote select membrane proteins interactions. The presence of specific CTD sequences have also been shown to promote specific associations (PMID: 16368689). The characterization of how individual ankyrins preferentially associate with specific subsets of client membrane proteins is beyond the scope of this work; however, the limitations of the proposed models should be discussed, especially in the context of the above-mentioned publications.

2) The description of mapping of AI-c for AnkB (exothermic to endothermic binding and biphasic binding in Figure 4—figure supplement 1) is difficult to follow and as described not very useful.

3) It would be useful if the authors could comment on autoinhibitory mechanisms in other repeat protein systems, and how they compare to the process they have uncovered in the ankyrins.

4) The authors should comment on how their results fit with the STORM imaging of ankG at axon initial segments together using antibodies directed against different domains of ankG. Leterrier et al., Cell Reports 2015.

5) Is it possible that the three AI domain interactions are cooperative? In other words, does affinity increase with two AI domains or three AI domains together?

6) Subsection “Features of the AnkB&G autoinhibition”, end of second paragraph. What evidence is there that alternative splicing doesn't affect the autoinhibitory mechanisms? Perhaps not the binding, but certainly the availability of the AI domains.

---

## [Author Response]

Essential revisions:1) This point concerns limitations of the model proposed by the authors. The authors propose a mechanism whereby binding of ankyrins to spectrins releases the AI domains allowing them to interact with their membrane binding targets. This model is not consistent with the fact that AnkG recruits bIV spectrin to the AIS, and that NF186 recruits ankG to nodes of Ranvier. Clustered ankG recruits spectrins to the AIS and nodes (see Yang et al., JCB), and clustering of ankG at the AIS does not require βIV spectrin (see Hedstrom et al., JCB) to cluster Na^+^ channels or NF186. While the model is plausible, the authors should discuss the limitations of the model. Moreover, the authors hypothesize that association of ankyrin with spectrin activates MBD associations with client membrane proteins. This hypothesis is attractive, but on its surface, seems to be in conflict with published data showing that full-length ankyrins purified from tissue can associate with high affinity to at least many of ankyrins' membrane protein clients in the absence of spectrin (e.g., PMID: 2452986, 3039371, 379653). The authors also hypothesize that phosphorylation events in the CTD may activate specific client associations by displacing CTD sequences blocking aspects of the MBD. This too is an attractive hypothesis and is one that has previously been suggested (e.g., PMID 2933395, 2970468). It may be that ankyrins are synthesized in a "closed" conformation, and upon interaction with spectrin, adopt a semi-open conformation that makes C-terminal sequences targets for phosphorylation. These phosphorylation events may then activate membrane protein binding activity. Ankyrins purified from tissues may have existent CTD phosphorylations that allow interaction with membrane proteins. As also discussed by the authors, the CTD is subject to extensive alternative splicing resulting in different ankyrins with different regulatory sequences. mRNA splicing events that eliminate specific blocking sequences may promote select membrane proteins interactions. The presence of specific CTD sequences have also been shown to promote specific associations (PMID: 16368689). The characterization of how individual ankyrins preferentially associate with specific subsets of client membrane proteins is beyond the scope of this work; however, the limitations of the proposed models should be discussed, especially in the context of the above-mentioned publications.

We thank the reviewers/editors for the wonderful comments and suggestions above. We fully agree with the reviewers/editors that our proposed model is not adequate and in several cases does not fit well with the literature observations of AnkG targeting and AnkG-mediated βIV spectrin and membrane protein recruitments at AIS/nodes of Ranvier. The likely scenario is that many regulatory mechanisms are used in a combinatorial fashion so that ankyrins can scaffold various target proteins in different cellular settings and in different tissues. The fully inhibited state of ankyrins with all three inhibitory segments (AI-a,b,c) occupying the ANK repeats may be an extreme state when ankyrins are synthesized and being trafficked to different cellular locations. Our proposed spectrin binding-induced ANK repeats opening, even if it is indeed employed in cells, is one of many regulatory mechanisms in activating ankyrins. We have revised our model and expanded the discussion about the model to include this critical point.

Alternative splicing is another way in regulating functions of ankyrins. Ankyrins at AIS/nodes of Ranvier are chiefly 270 kDa/480 kDa AnkG isoforms. Interestingly, these two larger AnkG isoforms do not contain the “AI-a” sequence in their CTDs due to alternative splicing. Thus, the 270 kDa/480 kDa AnkG seem to have different autoinhibition mechanisms with respect to other ankyrins (e.g. 190 kDa AnkG and 220 kDa AnkB). The lack of the “AI-a” site in 270 kDa/480kDa AnkG means that these two AnkG isoforms can still bind to Nav channels or NF186 (See Figure 2, AnkB 28-965; and also Hedstrom et al. JCB, 2008), albeit with much lower affinities (i.e. a form of semi-open conformation). Spectrin binding to the AnkG SBD may promote further opening of ANK repeats and thus leading to full engagements of the membrane targets of AnkG. Such mutual reinforcement of AnkG-mediated membrane microdomain assembly model is consistent with the findings showing that membrane targets and βIV spectrin mutually stabilize each other through AnkG at AIS or nodes of Ranvier (Yang et al., J neurosci 2004; Patzke et al., J Exp Med, 2016). Additionally, post-translational modifications (e.g. phosphorylation by different protein kinases) can provide another level of differential activations of ankyrins.

We have significantly revised the model and descriptions of the model to reflect the above points.

2) The description of mapping of AI-c for AnkB (exothermic to endothermic binding and biphasic binding in Figure 4—figure supplement 1) is difficult to follow and as described not very useful.

Thanks for the suggestion. We have removed the “Figure 4—figure supplement 1” and changed the corresponding descriptions in the revised manuscript.

3) It would be useful if the authors could comment on autoinhibitory mechanisms in other repeat protein systems, and how they compare to the process they have uncovered in the ankyrins.

We agree with the reviewers/editors’ suggestion and have modified the Discussion section to compare autoinhibition mechanisms of some other repeat-containing proteins. The combinatorial usage of multiple binding sites to engage their binding partners also exists in other elongated repeats containing proteins (Chook and Blobel, 2001; Conti et al., 1998; Graham et al., 2000; Huber and Weis, 2001; Kobe, 1999; Xu et al., 2010; Zhu et al., 2011). Whereas autoinhibition of an elongated repeats domain by multiple intrinsically discorded segments from the same protein seems to be much less common. A somewhat similar autoinhibition as ankyrins do is a karyopherin family protein importin-α. In that case, two NLS recognition sites on the armadillo repeats were shown to be autoinhibited by a sequence from importin-α’s N-terminal unstructured region (Catimel et al., 2001). It is noted that, unlike ankyrins, these two binding sites are inhibited by the exact same segment in a 1:2 stoichiometry instead of multiple segments with different sequences found in ankyrins. Except for ankyrin, only limited structures of autoinhibited repeats domains have been reported (PDB ID: 4KXF, 2BAP, 3VYC, 1IQ1 and 1IAL). Two of them are importin-α as mentioned above (1IQ1 and 1IAL); one is the armadillo repeats of Diaphanous-related formins (Drfs) autoinhibited by a short α helix from its distal C-terminal region (2BAP); one is the HEAT repeats of exportin chromosome region maintenance 1 (CRM1) autoinhibited by an immediate C-terminal α helix following the HEAT repeats (3VYC); and one is the nucleotide-binding and oligomerization domain-like receptor protein (NLRC4). The leucine-rich repeats (LRR) of NLRC4 is positioned to sterically occlude one side of NLRC4’s NBD domain and consequently sequester NLRC4 in a monomeric state. We have discussed these in the revised manuscript.

4) The authors should comment on how their results fit with the STORM imaging of ankG at axon initial segments together using antibodies directed against different domains of ankG. Leterrier et al., Cell Reports 2015.

Our data is consistent with the findings by Leterrier et al. They demonstrated that AnkG’s SBD exhibits a periodic pattern, while the C-terminus part of AnkG is not periodically arranged and is found ~32 nm radially deeper than the SBD in the axoplasm. AIS AnkG (270 kDa/480 kDa isoforms) does not contain AI-a on its CTs due to alternative splicing, so the major parts of CT of 270 kDa/480 kDa AnkG are not sequestered by MBD even if AI-b&c are in contact with MBD. As we have touched in our response to the comment #1, the conformation of 270 kDa/480 kDa AnkG at AIS/Nodes of Ranvier is likely open. As such, the signals of antibodies labeling the C-terminus part of AnkG should not appear near the membrane but is deeper into the axoplasm. Additionally, as the giant insertion regions and CTs of 270 kDa/480 kDa AnkG are largely unstructured, this probably explains why the CT is not periodically arranged in the STORM imaging experiments.

5) Is it possible that the three AI domain interactions are cooperative? In other words, does affinity increase with two AI domains or three AI domains together?

The AI domains in ankyrins are only partially cooperative. For example, the Kd of AI-b/MBD interaction is ~3.7 μM (Figure 3) and the Kd of AI-c/MBD interaction is ~0.36 μM (Figure 4); a linker sequence (AnkB 828-965) that contains both AI-b and AI-c binding to MBD with a Kd roughly 0.044 μM (Figure 2); If AI-b and AI-c are fully cooperative, the apparent Kd should be ~1 pM (3.7 μM × 0.36 μM ≈ 1.3 pM) instead of observed ~0.044 μM. The synergism between AI-a and AI-b is expected to be much weaker than that of AI-b and AI-c, because the spacing sequence between AI-a and AI-b segments is much longer than that of AI-b and AI-c (Figure 7). Such weak cooperativity among the inhibitory sites is advantageous for the combinatorial regulation model of ankyrins as we have described in our response to the comment #1. Though thermodynamically plausible, but proving such model by experiments is not an easy task.

6) Subsection “Features of the AnkB&G autoinhibition”, end of second paragraph. What evidence is there that alternative splicing doesn't affect the autoinhibitory mechanisms? Perhaps not the binding, but certainly the availability of the AI domains.

We searched the *Ensembl* (http://www.ensembl.org) database and found that in all the available splicing variant sequences of AnkB or AnkG (except for variants that without ANK repeats), both AI-b and AI-c are preserved in all variants. So, it seems that the sequences of AI-b and AI-c are not affected by alternative splicing in all AnkB/G variants. However, it is noted that the giant ankyrin-G isoforms (i.e. the 270 kDa/480 kDa AnkG) contain AI-b and AI-c in their linker regions but lose AI-a in their CTs due to alternative splicing. In contrast, the giant Ankyrin-B (i.e. 440 kDa AnkB) contains all three AI segments. Additionally, for AnkR, AI-a is missing in some splicing variants. We have revised the second paragraph of the subsection “Features of the AnkB&G autoinhibition” to make a clear description.